# Comparing Microscopic and Macroscopic Dynamics in a Paradigmatic Model of Glass-Forming Molecular Liquid

**DOI:** 10.3390/ijms23073556

**Published:** 2022-03-24

**Authors:** Giuseppe Porpora, Francesco Rusciano, Raffaele Pastore, Francesco Greco

**Affiliations:** Department of Chemical, Materials and Production Engineering, University of Naples Federico II, P.le Tecchio 80, 80125 Napoli, Italy; giuseppe.porpora@unina.it (G.P.); francesco.rusciano@unina.it (F.R.); francesco.greco@unina.it (F.G.)

**Keywords:** molecular glass-forming liquids, glass transition, molecular dynamics simulations

## Abstract

Glass transition is a most intriguing and long-standing open issue in the field of molecular liquids. From a macroscopic perspective, glass-forming systems display a dramatic slowing-down of the dynamics, with the inverse diffusion coefficient and the structural relaxation times increasing by orders of magnitude upon even modest supercooling. At the microscopic level, single-molecule motion becomes strongly intermittent, and can be conveniently described in terms of “cage-jump” events. In this work, we investigate a paradigmatic glass-forming liquid, the Kob–Andersen Lennard–Jones model, by means of Molecular Dynamics simulations, and compare the macroscopic and microscopic descriptions of its dynamics on approaching the glass-transition. We find that clear changes in the relations between macroscopic timescales and cage-jump quantities occur at the crossover temperature where Mode Coupling-like description starts failing. In fact, Continuous Time Random Walk and lattice model predictions based on cage-jump statistics are also violated below the crossover temperature, suggesting the onset of a qualitative change in cage-jump motion. Interestingly, we show that a fully microscopic relation linking cage-jump time- and length-scales instead holds throughout the investigated temperature range.

## 1. Introduction

When a liquid is cooled sufficiently fast below its melting temperature $T_{m}$, molecules do not have enough time to rearrange in an ordered structure and crystallization is avoided [1,2]. Liquids in these conditions are termed *supercooled*, and show a dramatic slowing-down of the dynamics as compared to that of a standard liquid, despite poor changes in their structure [2,3] (structural changes can be still detected using multi-point correlation functions and percolative approaches [4,5,6]). Macroscopic dynamical properties such as shear viscosity and diffusivity change indeed by several orders of magnitude upon supercooling [7]. As a matter of fact, below some conventional glass transition temperature $T_{g}<T_{m}$, the system eventually reaches a non-equilibrium disordered solid-like state, called *glass*, in which dynamics is arrested over the accessible timescales [1,8].

On a microscopic level, dynamics near the glass transition shows an intermittent single-particle motion, commonly known as Cage-Jump (CJ), with an alternation of localized vibrations inside the ‘cage’ created by the surrounding particles, and sudden ‘jumps’ to other cages [9,10,11,12,13]. While at high temperatures particles continuously overcome local energy barriers and smoothly change their neighbours, cage-jump dynamics becomes progressively more marked on lowering temperature, and is in fact clearly detectable in the supercooled state.

In the past few years, algorithms to characterize cage-jump events have been successfully developed, focusing either on fluctuations in single-particle trajectories [9,14,15,16], or on many-particles rearrangement [17,18,19], or on transitions in the energy landscape [20,21]. The statistics obtained through these algorithms, such as cage-duration and jump-length distributions, can then be used as input in simple models of glassy dynamics, like for example the Continuous Time Random Walk (CTRW) model [22,23].

When dealing with cage-jumps, commonly measured quantities include the *caging time* (or *exchange time*) $t_{c}$, i.e., the time between two successive jumps, the *persistence time*
$t_{p}$, i.e., the time for a particle to perform the first jump (with respect to an arbitrarily chosen t=0), and the *jump length*
$\Delta r_{J}$, i.e., the distance between two successive cages. The averages of these quantities identify the main time- and length-scales of the microscopic intermittent dynamics [11].

The behaviour of such time- and length-scales helps to rationalize several aspects of the macroscopic dynamics. For example, a successful application stands in the identification of a *microscopic* counterpart of the so-called Stokes–Einstein Breakdown (SEB), which is a common hallmark of many glass-formers [24,25,26]. In these materials, the two *macroscopic* timescales for diffusion and structural relaxation, $\tau_{D}$, and $\tau_{\alpha }$, respectively, clearly decouple upon cooling, thus violating the celebrated Stokes–Einstein relation $\frac{\tau_{\alpha }}{\tau_{D}}=const$. This behaviour is found to be mirrored on a microscopic ground by the decoupling of the two fundamental cage-jump timescales, the average caging time $\langle t_{c}\rangle$ and the persistence time $\langle t_{p}\rangle$, as reported in a number of numerical studies [27,28,29].

For the sake of clarity, we specify that we here call “macroscopic” a quantity that can be obtained from a bulk measurement, like the structural relaxation time and the diffusion coefficient, which can be measured, for example, through scattering techniques. Conversely, we call “microscopic” a quantity that can be only obtained by resolving the single-particle dynamics (i.e., by recording single-particle trajectories). The latter is indeed the case for cage-jump quantities, which are measured by segmenting single particle trajectories. From an experimental perspective, this implies that our “microscopic quantities” are hardly measured in molecular liquids. As a matter of fact, while single-particle motion can be readily monitored in experiments on colloidal model-systems, simulations have long been the only way to follow single-particle trajectories in molecular systems. Only recently, important advances in the techniques of single-molecule imaging (e.g., single-molecule fluorescence microscopy) may provide an alternative to simulations [30,31,32,33,34,35]: collecting a sufficiently large number of sufficiently long-lasting trajectories, however, is still a limiting factor in experiments [33]. For our specific purpose, for example, large ensembles of long trajectories are necessary to fairly sample the tails of the $t_{c}$ and $t_{p}$ distributions, and to reliably estimate their averages.

In this work, we perform a comparative study on the characteristic scales of the microscopic cage-jump motion and of the macroscopic dynamics in a paradigmatic model of a molecular glass-forming liquid. To this aim we investigate, via Molecular Dynamics (MD) simulations, the popular Kob–Andersen Lennard–Jones binary mixture (KALJ) [3], and identify cage-jumps trough application of an established algorithm [14]. Results point to the existence of two regimes on progressive cooling. At relatively high temperature, the decoupling between $\tau_{D}$ and $\tau_{\alpha }$ (SEB) and between $\langle t_{c}\rangle$ and $\langle t_{p}\rangle$ is a modest one, and predictions from Mode Coupling fits, CTRW and “lattice-glass” models [27,28,36] (also known as Kinetically Constrained Models [37]), drawing on the identified CJs, seem satisfactory. On further supercooling, by contrast, strong decouplings take place starting from the same temperature, with the microscopic one being much steeper. In the same low-temperature range, deviations from the aforementioned predictions become apparent. Intriguingly, the crossover between the two regimes does not affect the temperature-dependence of the average square jump length $\langle \Delta r_{J}^{2}\rangle$. Similarly, we find that a fully microscopic relation among CJ time and length scales, $\frac{\langle t_{p}\rangle }{\langle t_{c}\rangle }\propto {\langle \Delta r_{J}^{2}\rangle }^{-1}$, recently reported for other glass-formers [38], holds in good approximation throughout the investigated temperature range.

## 2. Materials and Methods

### 2.1. Simulations

To obtain the microscopic observables, we performed NVT molecular dynamics simulations in LAMMPS [39] of a standard Kob–Andersen 80:20 (A:B) binary Lennard–Jones mixture (KALJ) [3] in the range of temperature T=[0.445:0.6], and apply the cage-jump algorithm described below. The simulated system is made by a total number $N=10^{3}$ particles, at number density ρ=1.1998, as in Ref. [40] (in the same work, it was checked that the slight difference with the number density ρ=1.204 originally used by Kob and Andersen [41] has negligible effects on the dynamics). Particles of species *i* and *j* interact via a Lennard–Jones potential with energy scale $\epsilon_{ij}$ and length scale $\sigma_{ij}$. All particles have the same mass *m*. As commonly done in molecular dynamic simulations, units are reduced so that $\sigma_{AA}=\epsilon_{AA}=m=k_{B}=1$ ($k_{B}$ is the Boltzmann constant), which implies that the time is expressed in unit of $\sqrt{\frac{m}{\epsilon_{AA}}}\sigma_{AA}$. Such molecular dynamic unit will be generically indicated with the symbol mdu in the following figures. The other values of the parameters are set as follows: $\epsilon_{AB}=1.5$; $\sigma_{AB}=0.8$; $\epsilon_{BB}=0.5$; $\sigma_{BB}=0.88$.

On the other hand, the relaxation and diffusive times plotted in this work are taken from Ref. [40] and cover a wider range of temperatures, T = 0.39–0.7. These simulations were performed using the recently introduced Parallel Tempering protocol, also know as “swap” dynamics [42], which enables system equilibration down to very low temperatures as compared to standard simulations.

All data presented in this work refer to A-type (small) particles in Kob–Andersen mixture. At all considered temperatures, supercooled liquids are at equilibrium condition and, therefore, above any reasonable definition of the glass transition temperature $T_{g}$.

### 2.2. Cage-Jump Algorithm and Microscopic Observables

The statistical features of the intermittent dynamics have been investigated using the cage–jump algorithm introduced in Ref. [14]. We associate to each particle, at each time *t*, the fluctuations $S^{2}\left(t\right)$ of its position computed over the interval $\left[t-10t_{b}:t+10t_{b}\right]$, with $t_{b}$ being the ballistic timescale. The trajectory of each particle is then segmented in cages and jumps, considering a particle to be in a cage at time *t* if $S^{2}\left(t\right)$ is smaller than the Debye–Waller factor $u^{2}\left(T\right)$, defined as in Ref. [43]. Otherwise, the particle is considered to be jumping. This procedure gives access to the caging time $t_{c}$, the persistence time $t_{p}$ and the jump length $\Delta r_{j}$. Microscopic observables $\langle t_{c}\rangle$, $\langle t_{p}\rangle$ and $\langle \Delta r_{j}^{2}\rangle$ are then computed by averaging over all segmented trajectories.

### 2.3. Macroscopic Observables

The structural relaxation time $\tau_{\alpha }$ is obtained from the Intermediate Self Scattering Function, defined as
$$F_{s}\left(q,t\right)=\frac{1}{N}\sum_{j}^{N}\langle e^{-iq\cdot [r_{j}\left(\Delta t\right)-r_{j}\left(0\right)]}\rangle$$
where 〈·〉 denotes average performed over time origin, q is the probing wave-vector and q=|q|. In particular, $\tau_{\alpha }$ is defined as the time at which $F_{s}\left(q^{*},t\right)$ reaches an arbitrary threshold of 1/e, with $q^{*}=7.25$ being the wave-vector corresponding to the first peak of the static structure factor S(q) [44].

The diffusion timescale $\tau_{D}$ represents the average time for a particle to diffuse over the length scale of its diameter. Precisely, it is defined as $\tau_{D}=\frac{\sigma_{AA}^{2}}{6D}$, *D* being the diffusion constant estimated from a linear fit $\langle r^{2}\left(t\right)\rangle =6Dt$ to the long-time Fickian regime of the mean square displacement.

## 3. Results

We start our investigation by showing, in Figure 1a, the relevant timescales (both microscopic and macroscopic) as a function of temperature *T*. As stated in Section 2, the microscopic times, $\langle t_{p}\rangle$ and $\langle t_{c}\rangle$, are extracted through CJ algorithm from MD simulations performed in the range T = 0.445–0.6; Macroscopic datasets, instead, are taken from a work by Coslovich and coworkers [40], who explored an unprecedentedly broad range of temperature, T = 0.39–0.7. The macroscopic timescales, $\tau_{\alpha }$ and $\tau_{D}$ are defined from the self scattering function $F_{s}$ and the diffusion coefficient, respectively, (see Section 2).

Figure 1a shows that $\tau_{\alpha }$ is the most steeply growing timescale upon cooling, while $\langle t_{c}\rangle$ is the least increasing one. In the figure, it should also be noticed that the range covered by the macroscopic timescales extends well below the “effective” critical temperature of dynamical arrest $T_{c}^{KA}=0.435$ estimated by Kob–Andersen [3] and confirmed in later works (e.g., Ref. [45]). Such an effective temperature was obtained by means of fits to the data inspired by Mode Coupling Theory (MCT), i.e., through power-laws of $T-T_{c}^{KA}$ with sligthly different exponents for $\tau_{\alpha }$ and $\tau_{D}$. However, it is worth remarking that ideal MCT would return a significantly higher critical temperature and a unique power-law exponent for those two timescales. As a matter of fact, data in Figure 1a obey an MCT-like trends (dashed lines) for temperatures higher than about T=0.47; at lower temperatures, instead, deviations from this behaviour are observed. Both timescales increase slower than power-law prediction, with no hint to a finite temperature divergence, consistently with other studies [46,47]. Below T=0.47, $\tau_{\alpha }$ remains lower but increases pretty faster than $\tau_{D}$, with the two timescales closely crossing at very low *T*. Thus, the present datasets of macroscopic timescales include a crossover between two distinct temperature regimes.

Microscopic data actually cover a range of temperatures that entirely lies above $T_{c}^{KA}$. Yet, the investigated temperature range fully encompasses the just mentioned crossover around T=0.47. Indeed. CJ timescales data also show a signature of the same crossover: $\langle t_{c}\rangle$ and $\langle t_{p}\rangle$ closely coincide at “high” temperatures (they would exactly coincide in a standard Brownian motion), and sharply decouple at “low” temperatures. Such a decoupling can be rationalized by considering that short waiting times, corresponding to the “fast” particles performing many jumps in a short time interval, have a major impact on $\langle t_{c}\rangle$, but poorly affect $\langle t_{p}\rangle$. Indeed, $\langle t_{p}\rangle$ is obtained by averaging only over the waiting times before the first jump of each particle, and is therefore more sensitive to the “slow” particles. We notice that the decoupling between $\langle t_{p}\rangle$ and $\langle t_{c}\rangle$ resembles the decoupling between the characteristic relaxation times of the late and early relaxation processes, also known as the α and the β (or Johary–Goldstein) relaxations [1]. This, in turn, would suggest a connection between $\langle t_{p}\rangle$ and $\tau_{\alpha }$ and between $\langle t_{c}\rangle$ and $\tau_{\beta }$. At a qualitative level, the connection between $\tau_{\alpha }$ and $\langle t_{p}\rangle$ is ascribed to the fact that both times are controlled by the “last” particles leaving their original cages [14,27,28,48]; this connection will be quantitatively tested here. On the other hand, the link between $\langle t_{c}\rangle$ and $\tau_{\beta }$ is more elusive. As a matter of fact, the just mentioned similarities between decouplings suggest that there is not a clear-cut separation between the β-relaxation time (i.e., when particles start “feeling” the constraint of their cages) and the total caging time of the fast particles.

Turning to the spatial feature of cage-jumps. in Figure 1b we report the mean square jump length $\langle \Delta r_{J}^{2}\rangle$ as a function of temperature. $\langle \Delta r_{J}^{2}\rangle$ is found to decrease roughly linearly by a factor 5 on lowering the temperature. For this quantity, then, there is no sign of the crossover.

To analyze similarities and differences between microscopic and macroscopic timescales, we start with a comparison of the temperature dependence of the two ratios $\frac{\tau_{\alpha }}{\tau_{D}}$ and $\frac{\langle t_{p}\rangle }{\langle t_{c}\rangle }$ (Figure 2). Both ratios, normalized here by their value at T=0.6, show a modest increase while in the high temperature regime, and a sharp growth below the crossover. However, down to T=0.445 (the lowest available temperature for microscopic timescales), $\frac{\langle t_{p}\rangle }{\langle t_{c}\rangle }$ increases approximately by factor 6, whereas $\frac{\tau_{\alpha }}{\tau_{D}}$ goes only up to 2. For the macroscopic ratio to attain the six-fold increase of the microscopic ratio, it is necessary to go down to temperature as small as T=0.39 (the smallest available temperature even exploiting “swap” dynamics [40]).

Thus, the decoupling of $\langle t_{p}\rangle$ and $\langle t_{c}\rangle$ is not only a proxy of macroscopic SEB, as suggested elsewhere [27,28,29,38,48,49,50,51], but also a precursor of this phenomenon, being already clearly detectable just below the crossover temperature.

To catch a further evidence of the crossover, we show in Figure 2b a scatter plot of the two ratios in the temperature regime where both of them are available. Two regions can be readily distinguished in the figure, corresponding in fact to the two aforementioned temperature regimes. Below the crossover, the scatter plot apparently shows a linear increase. High temperature data are still compatible with a linear increase, although with a much larger slope. Of course, inferring a trend in the high-temperature regime is less robust, just because the two ratios remain there always close to unity.

To further explore the connections between macroscopic and microscopic CJ dynamics, and how those are affected by the crossover, we now test some predictions from CTRW and lattice models. We plot in Figure 3 $\tau_{D}$ vs. $\frac{\langle t_{c}\rangle }{\langle \Delta r_{J}^{2}\rangle }$. In the high temperature regime, the trend of the data is well captured by a linear fit (dashed line), which agrees with the prediction $\tau_{D}\propto \frac{\langle t_{c}\rangle }{\langle \Delta r_{J}^{2}\rangle }$ of the standard CTRW model [23]. Of course, we have implicitly assumed that $t_{c}$ and $\Delta r_{j}$, as measured with our CJ algorithm, play the role of the exchange time and the step size in the CTRW model.

Below the crossover temperature, i.e., for the highest points in the figure, deviations from the CTRW prediction appear, with the measured $\tau_{D}$ increasingly exceeding the theoretical values.

Passing to a comparison with lattice models, we notice that the temporal statistics of jumps will play the role of the temporal statistics of lattice steps. On the other hand, the variable $\Delta r_{J}$ cannot be considered at all, of course, since this quantity is “by construction” a constant in lattice models. In spite of this, the comparison of our data with lattice models confirms the emerging scenario. Both the predictions $\tau_{D}\propto \langle t_{c}\rangle$ and $\tau_{\alpha }\propto \langle t_{p}\rangle$ [27,28,36], tested in Figure 4a,b, respectively, are well obeyed in the high-temperature regime, while significant deviations are observed below the crossover temperature. Also in this case, the predictions underestimate the macroscopic timescales.

Finally, in Figure 5, we test an interesting result, recently obtained for different glass-forming liquids [38], namely, the linear relation $\frac{\langle t_{p}\rangle }{\langle t_{c}\rangle }\propto {\langle \Delta r_{J}^{2}\rangle }^{-1}$, providing a connection among the three CJ microscopic quantities. We do find that this fully microscopic relation works very well also for the 3d KALJ liquid investigated here. Interestingly, the robustness of this relation is not affected by the temperature crossover, except for minor deviations at the very highest temperatures.

## 4. Discussion

In this work, we made a comparative study of microscopic CJ motion and macroscopic dynamics of a paradigmatic model of molecular glass-former, namely the 3d KALJ liquid.

The overall behaviour of time and length scales show the generic features expected for glass-forming systems [16,28,29,48].

Both couples of macroscopic and microscopic timescales markedly separate upon cooling, and especially do so below a crossover temperature T≃0.47, which then discriminates between two distinct regimes. Interestingly, such crossover temperature also coincides with the onset of deviations from MCT-like behaviour. We find that, at variance with other systems, in which the two ratios $\frac{\tau_{\alpha }}{\tau_{D}}$ and $\frac{\langle t_{p}\rangle }{\langle t_{c}\rangle }$ are linearly related on decreasing temperature [38], in the present case the microscopic ratio actually increases quite faster than the macroscopic one when the low-temperature regime is entered. Hence, the decoupling of the CJ timescales $\langle t_{p}\rangle$ and $\langle t_{c}\rangle$ in 3d KALJ liquid not only is a proxy of the macroscopic SEB, but can also be seen as an “early-warning” (in temperature) of its occurrence. We further find that, above the crossover temperature, CTRW and lattice-glass predictions relating the macroscopic and microscopic scales fairly-well describe our data. By contrast, those predictions are violated in the low-temperature regime. Such deviations may perhaps be ascribed to the emergence of correlations between successive jumps of a particle (e.g., back and forward movements), as suggested elsewhere [50], which would be rich of implications. For example, the identification of correlated jumps would prevent a direct mapping with the steps of a CTRW. We may also notice here that the presence of anti-correlated jumps in the statistics (a sort of “false positive event”) would obviously lead to an underestimate of the average CJ times, which would explain the discrepancy shown in Figure 3.

Finally, we find a linear relation between $\frac{\langle t_{p}\rangle }{\langle t_{c}\rangle }$ and ${\langle \Delta r_{j}^{2}\rangle }^{-1}$, which suggests the existence of a correlation between the fundamental length and time scales of CJ motion. Noticeably, this feature appears also in other glass-forming liquids [38]. It is interesting to underline that this fully microscopic relation seems to be unaffected by the crossover between the the two temperature regimes. Similarly, the temperature dependence of the microscopic length-scale $\langle \Delta r_{J}^{2}\rangle$ does not show any hint of the crossover.

As for perspectives, it would be interesting to elucidate the origin of the fully microscopic relation $\frac{\langle t_{p}\rangle }{\langle t_{c}\rangle }\propto {\langle \Delta r_{j}^{2}\rangle }^{-1}$, which at present is not understood on a theoretical background. Other main outlooks include studying the relevance of jump correlations occurring in the low temperature regime, and the possibility to define CTRW-like jumps over a wider temperature range. Finally, we would like to emphasize that recent progresses of single-molecules imaging may open the way to an experimental study of the cage-jump motion in molecular supercooled liquids [30,31,32,33,34,35], provided that satisfactory trajectory ensembles could be collected. Such advanced experiments are, of course, on demand to complement and validate the picture emerging from molecular dynamics simulations.

## Figures and Tables

**Figure 1 ijms-23-03556-f001:**
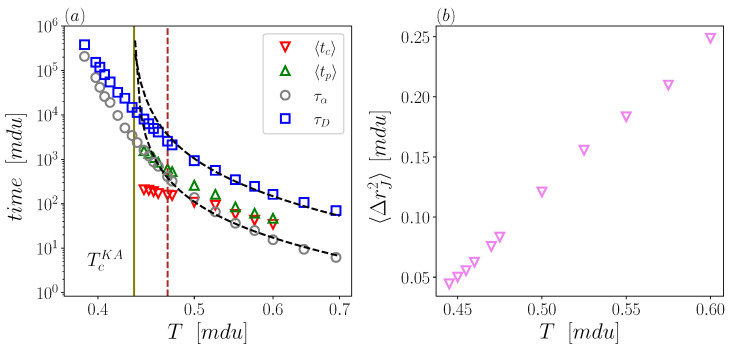
(**a**) Macroscopic ($\tau_{\alpha }$ and $\tau_{D}$) and microscopic ($\langle t_{p}\rangle$ and $\langle t_{c}\rangle$) timescales as a function of temperature. Dashed lines are MCT-like fits ${\left(T-T_{c}^{KA}\right)}^{-\gamma }$ with $T_{c}^{KA}=0.435$, γ=2.2 for $\tau_{\alpha }$ and γ=2.0 for $\tau_{D}$. Fitting parameters are taken from [3,44]. Green vertical solid line represents the critical temperature $T_{c}^{KA}$, red vertical dashed line marks T=0.47. (**b**) Mean square jump length $\langle \Delta r_{J}^{2}\rangle$ as a function of temperature.

**Figure 2 ijms-23-03556-f002:**
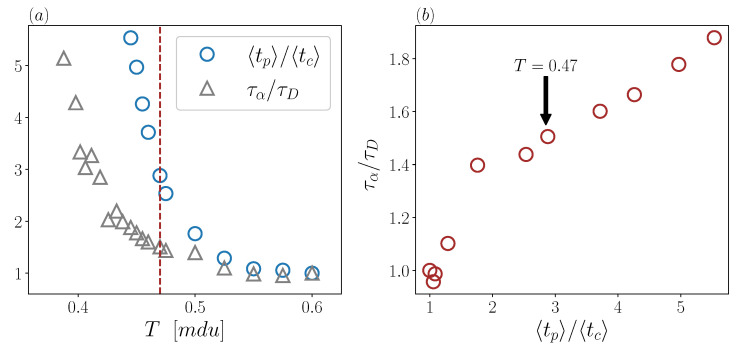
(**a**) $\frac{\tau_{\alpha }}{\tau_{D}}$ and $\frac{\langle t_{p}\rangle }{\langle t_{c}\rangle }$ as a function of temperature. Ratios of microscopic and macroscopic timescales have been divided by their own value at T=0.6, which is the highest available temperature for the CJ dataset. Red vertical line indicates T=0.47 (**b**) Scatter plot of the macroscopic versus microscopic timescales ratios, rescaled as in panel (**a**).

**Figure 3 ijms-23-03556-f003:**
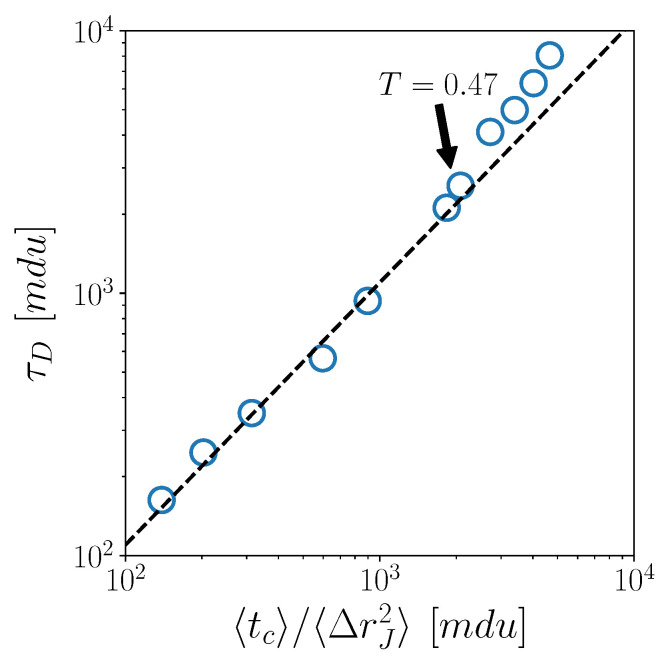
Scatter plot of $\tau_{D}$ versus $\frac{\langle t_{c}\rangle }{\langle \Delta r_{J}^{2}\rangle }$. Dashed black line represents a linear fit, corresponding to the CTRW prediction.

**Figure 4 ijms-23-03556-f004:**
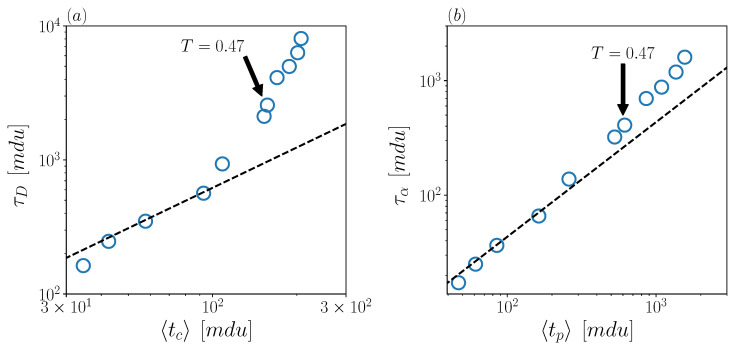
Scatter plots of (**a**) $\tau_{D}$ versus $\langle t_{c}\rangle$ and (**b**) $\tau_{\alpha }$ versus $\langle t_{p}\rangle$. Dashed black line represents a linear fit, corresponding to the lattice-model predictions.

**Figure 5 ijms-23-03556-f005:**
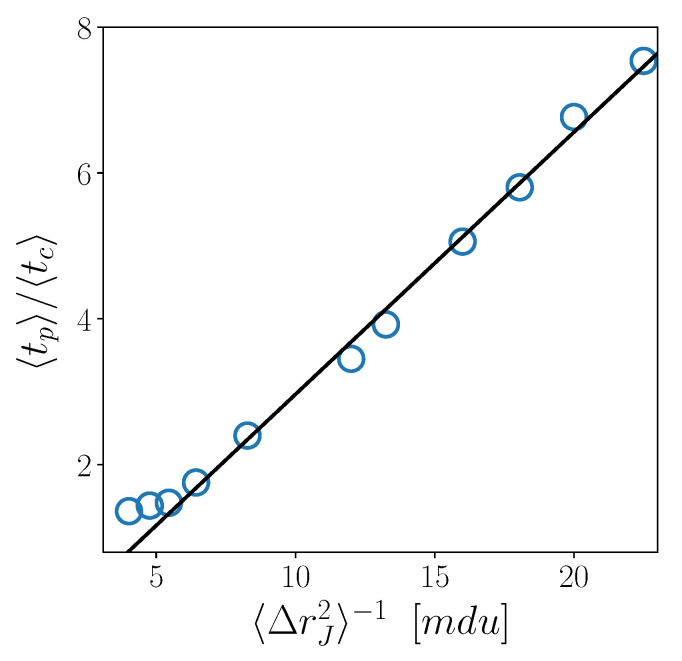
Scatter plot of $\frac{\langle t_{p}\rangle }{\langle t_{c}\rangle }$ versus ${\langle \Delta r_{j}^{2}\rangle }^{-1}$. Solid line represents a linear fit.

## Data Availability

The data that support the findings of this study are available from the corresponding author upon reasonable request.

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
