# Peer review of "Comparing Microscopic and Macroscopic Dynamics in a Paradigmatic Model of Glass-Forming Molecular Liquid"

_ijms, 2022, doi:10.3390/ijms23073556_

Round 1

Reviewer 1 Report

The manuscript by Porpora et al. is devoted to a still challenging problem of glass-forming liquids, which is the impact of microscopic molecular dynamics on the macroscopic dynamic quantities, the latter of which can be much easier determined experimentally. By using molecular dynamics simulations in the Kob-Andersen binary Lennard-Jones model, which is a prototypical model of glass-forming liquids, the authors continue their previous research relied on other simulation models. In this way, they investigate the potential correlations between the microscopic and macroscopic quantities, where the former ones are the average caging time between two successive jumps, the persistence time, and the average square jump length as well as the latter ones are the structural relaxation time and the diffusion coefficient expressed in the form of the diffusion time. Considering predictions based on different theoretical models and results of their analyses, the authors conclude that the decoupling of the mean persistence time and the mean caging time is a precursor of the Stokes-Einstein breakdown, i.e., the decoupling between the structural relaxation time and the diffusion coefficient, which is often observed in case of molecular glass-formers. This finding is definitely worthy of publication in the International Journal of Molecular Sciences. However, the authors should address a few minor points listed below before the final approval of the manuscript for publication.

  1. In Introduction, the authors wrote that “From an experimental perspective, this implies that our "microscopic quantities” cannot be obtained in molecular liquids, since this would require resolving the dynamics at the atomic scale.“ and clarified that “… we call “microscopic” a quantity that can be only obtained by resolving the single-particle dynamics (i.e. by recording single-particle trajectories)”. These sentences may give an impression that the measurements of single-particle molecular dynamics have never been performed, while such experiments have been carrying out for several years despite they are difficult. For instance, one can invoke Laura Kaufman’s works, including paper in J. Phys. Chem. Lett. 2, 438 (2011), and a paper by Yukimoto et al. [Sci. Rep. 3, 1855 (2013)]. Thus, the authors should compare the novel experimental techniques to the molecular dynamics simulation method in Introduction, and then inform the readers whether the caging time and the persistence time are still impossible to be gathered from measurements. Of course, the potential possibility of the experimental determination of these quantities would not exclude the MD simulation studies, but enabled a comparison of the experimental and simulation results.

  2. In the manuscript, it is difficult to find a density value that has been set in the MD simulations in the NVT ensemble. One can guess that it equals 1.204, that is the same as in Refs. 31 and 34. The readers should not be forced to search for such a basic parameter in cited papers.

  3. Finally, in the context of the manuscript outcome, it would be interesting to know whether the authors could discern a linkage between the mean caging time and the Johari-Goldstein secondary relaxation time. The latter is usually interpreted as a reflection of the small-angle rotation of entire molecules and it is often considered as a precursor of the structural relaxation. Of course, it is obvious that one cannot investigate any rotations in the KABLJ model. However, the nice result shown in Fig. 1 that the mean persistence time increases similarly as the structural relaxation time on decreasing temperature, while the mean caging time decouples from the persistence and structural relaxation times at a crossover temperature, reveals a similarity to an analogous decoupling between the structural relaxation and the Johari-Goldstein secondary relaxation in the glass-forming liquids. Is it only a coincidence? Maybe this remark will inspire the authors to discuss briefly this issue in the revised manuscript.   

Reviewer 2 Report

This manuscript does not correspond to the high-quality criteria of the journal of IJMS. Therefore, I recommend rejecting this manuscript. The novelty level of this manuscript is very low. Although it is adequately written, it offers no critical information and no new slant on the review topic. Most of the content in the manuscript is better for the journal with a lower level. The introduction section should be improved. Characterizations tests are mandatory and need to verification and validation of the simulation model. Interpretation of analyses is very poor and requires re-analysis and further testing.

Author Response

response

Reviewer 3 Report

This is a good quality work where the authors have used MD simulation to investigate the Kob-Andersen Lennard-Jones binary mixture comparing the macroscopic and microscopic descriptions of its dynamics on approaching the glass-transition. Authors have identified clear changes in the relations between macroscopic timescales and cage-jump quantities which occur at the crossover temperature where Mode Coupling-like description starts failing. Although authors do not note any of the works which do also clearly indicate structural changes at the glass transition (see e.g. doi:10.1021/acs.jpcb.0c00214 and references there) the manuscript needs just few amendments and can be published.

Some minor type amendments shall be done accounting for the following comments:

Chapter 2.1, lines 92 and 100: explain the units used for temperature, line 98 – are the units of energy, mass and length atomic units?

Figure 1: Give explanation of the “mdu” units used and explain why they are the same for both time and temperature (a) and both square length and temperature (b).

Figure 2: Do authors relate any of the T shown to the temperature of glass transition?
